# High-Risk HPV Screening Initiative in Kosovo—A Way to Optimize HPV Vaccination for Cervical Cancer

**DOI:** 10.3390/diseases12080189

**Published:** 2024-08-19

**Authors:** Jessica L. Bentz, Rachael E. Barney, Natalia Georgantzoglou, Suzana Manxhuka-Kerliu, Vlora Ademi Ibishi, Brikene Dacaj-Elshani, Suyapa Bejarano, Paul E. Palumbo, Arvind Suresh, Ethan P. M. LaRochelle, William P. Keegan, Teresa L. Wilson, Betty J. Dokus, Kenneth C. Hershberger, Torrey L. Gallagher, Samantha F. Allen, Scott M. Palisoul, Heather B. Steinmetz, Linda S. Kennedy, Gregory J. Tsongalis

**Affiliations:** 1Dartmouth Hitchcock Medical Center, Department of Pathology and Laboratory Medicine, Lebanon, NH 03756, USA; rachael.e.barney@hitchcock.org (R.E.B.); natalia.georgantzoglou@gmail.com (N.G.); ethanlarochelle@gmail.com (E.P.M.L.); william.p.keegan@gmail.com (W.P.K.); teresa.l.wilson@hitchcock.org (T.L.W.); betty.j.dokus@hitchcock.org (B.J.D.); hershbergerkc@gmail.com (K.C.H.); torrey.l.gallagher@hitchcock.org (T.L.G.); samantha.f.allen@hitchcock.org (S.F.A.); scott.m.palisoul@hitchcock.org (S.M.P.); heather.b.steinmetz@hitchcock.org (H.B.S.); gregory.j.tsongalis@hitchcock.org (G.J.T.); 2Theodore and Audrey Geisel School of Medicine, Dartmouth College, Hanover, NH 03755, USA; 3Faculty of Medicine, University of Prishtina, 10000 Prishtina, Kosovo; suzana.kerliu@uni-pr.edu (S.M.-K.); vlora.ibishi@uni-pr.edu (V.A.I.); brikena.elshani@uni-pr.edu (B.D.-E.); 4La Liga Contre del Cancer, San Pedro Sula 21104, Honduras; sube.mayanet@gmail.com; 5Dartmouth Hitchcock Medical Center, Department of Medicine, Lebanon, NH 03756, USA; paul.e.palumbo@dartmouth.edu; 6Dartmouth Cancer Center, Lebanon, NH 03756, USA; arvind.suresh@hitchcock.org (A.S.); lindaskeweskennedy@gmail.com (L.S.K.)

**Keywords:** human papillomavirus, high risk HPV, cervical cancer screening, cervical cancer

## Abstract

Nearly all cervical cancers are caused by persistent high-risk human papillomavirus (hrHPV) infection. There are 14 recognized hrHPV genotypes (HPV 16, 18, 31, 33, 35, 39, 45, 51, 52, 56, 58, 59, 66, and 68), and hrHPV genotypes 16 and 18 comprise approximately 66% of all cases worldwide. An additional 15% of cervical cancers are caused by hrHPV genotypes 31, 33, 45, 52, and 58. Screening patients for hrHPV as a mechanism for implementation of early treatment is a proven strategy for decreasing the incidence of HPV-related neoplasia, cervical cancer in particular. Here, we present population data from an HPV screening initiative in Kosovo designed to better understand the prevalence of the country’s HPV burden and local incidence of cervical cancer by hrHPV genotype. Nearly 2000 women were screened for hrHPV using a real-time polymerase chain reaction (real-time PCR) assay followed by melt curve analysis to establish the prevalence of hrHPV in Kosovo. Additionally, DNA was extracted from 200 formalin-fixed, paraffin embedded cervical tumors and tested for hrHPV using the same method. Cervical screening samples revealed a high prevalence of hrHPV genotypes 16 and 51, while cervical cancer specimens predominantly harbored genotypes 16, 18, and 45. This is the first comprehensive screening study for evaluating the prevalence of hrHPV genotypes in Kosovo on screening cervical brush samples and cervical neoplasms. Given the geographic distribution of hrHPV genotypes and the WHO’s global initiative to eliminate cervical cancer, this study can support and direct vaccination efforts to cover highly prevalent hrHPV genotypes in Kosovo’s at-risk population.

## 1. Introduction

Globally, HPV-related cervical carcinomas account for the fourth leading cause of death in women, with the highest incidence in low human development index (HDI) countries [1,2]. The majority of cervical cancers are caused by a persistence of hrHPV genotypes (HPV 16, 18, 31, 33, 35, 39, 45, 51, 52, 56, 58, 59, 66, and 68), and hrHPV genotypes 16 and 18 comprise approximately 66% of all cases worldwide [3,4,5,6,7]. An additional 15% of cervical cancers are caused by hrHPV genotypes 31, 33, 45, 52, and 58 [7]. In low-and middle-income countries (LMICs), cervical cancer is the primary cause of cancer-related mortality. Cervical neoplasms are frequently identified at an advanced stage due to lack of screening, limited treatment options both geographically and financially, cultural restrictions on women accessing care, and higher rates of concomitant immunosuppressive conditions, including HIV [8,9]. Over more than five decades, high-income countries with less restricted access to health-related resources have greatly reduced death from cervical cancer by widespread development and adoption of screening initiatives, including the Pap test, advanced hrHPV detection assays, and vaccination strategies [10,11]. Conversely, lesser-resourced countries face significant challenges with the implementation of cervical cytology screening and vaccination due to a lack of adequate infrastructure, training and personnel, health education programs for the population, and funding [12,13,14,15].

Two commercially available FDA-approved vaccines can protect against some of the most common types of cancer-causing high-risk HPV (hrHPV); one of the vaccines is funded by the GAVI alliance (GAVI). Just 18 low-income countries around the world utilize GAVI funding to vaccinate young girls against HPV [16]. The GAVI-funded vaccine is a less expensive but more narrowly protective quadrivalent vaccine against low-risk HPV (lrHPV) types 6 and 11 and hrHPV types 16 and 18. The relatively more expensive (non-GAVI covered) vaccine is a broader coverage nonavalent vaccine against lrHPV types 6 and 11 and hrHPV types 16, 18, 31, 33, 45, 52, and 58 [16,17]. In LMICs, adult women generally do not have access to HPV vaccination, except by personal payment at private clinics. 

Furthermore, numerous large scale studies have shown a markedly varied geographic distribution of hrHPV genotypes across the globe in both cervical cytology specimens and cervical carcinomas [18,19,20]. What is most prevalent in one location is not true for all. This information is an important component to inform design and implementation of an optimized HPV vaccination program based on which hrHPV types are most prevalent in a particular country or region’s population. At the time our study was designed, data about HPV distribution, by type, were not available for some LMICs. Our target was Kosovo, which was selected because there were no related published data on the country, and pathologists working in Kosovo had no population health data in the pipeline. The overall estimated prevalence of HPV in Kosovo is 13.1% [21,22]. Further, knowledge about HPV’s association with cervical cancer and the HPV vaccine was limited outside of urban regions in the country [21]. In addition to Kosovo being understudied, our university had a long-standing relationship with the University of Prishtina based on training exchanges and significant partnerships to assist Prishtina University after the war. Our proposed approach met the unique needs of Dartmouth, which sought a fresh site to assist with obtaining baseline research, and the University of Prishtina, which needed knowledge about the hrHPV genotypes causing cervical cancer in local women as a means of customizing prevention, screening, and treatment plans in the primary health care system. Our jointly planned study examined the prevalence of hrHPV genotypes in screening samples on cervical brushes and paraffin blocks of cervical cancer tumors obtained from the University of Prishtina (Prishtina, Kosovo). These data could potentially inform local public health policy makers concerned with optimizing population-based HPV screening, since a national HPV co-testing program was not in place and vaccination programs were still in the early phases of implementation. 

## 2. Materials and Methods

The studies described in this manuscript were approved by the Dartmouth Health Institutional Review Board (Study#02000648, Lebanon, NH, USA) and the University of Prishtina (Study#259, Prishtina, Kosovo). To prospectively collect the cervical specimens in Kosovo, two professors of obstetrics and gynecology with medical residents obtained ongoing Pap smears from patients being seen in an obstetrics and gynecology clinic that is part of the University Clinical Center of Kosovo. All women with atypical cytology results from their examination were referred for appropriate follow-up care at the University Clinical Center.

After consenting to study participation, de-identified cervical brush specimens were collected from 1994 asymptomatic women in Kosovo by trained clinicians and medical residents. A sterile smear brush (MEDBAR TIBBI MALZEMELER TURIZM SANAYI VE TICARET LTD STI) was used to collect a cervical specimen, which was then placed into a 50 mL conical tube with a cap and stored at room temperature before shipping to the Dartmouth Health Department of Pathology and Laboratory Medicine via FedEx.

Cervical tumor tissues in paraffin blocks (*n* = 200) were retrieved from the pathology archives in Kosovo and reviewed by a pathologist, then shipped to the Dartmouth Health Department of Pathology and Laboratory Medicine at room temperature via FedEx courier. Four formalin-fixed, paraffin-embedded (FFPE) rolls were cut at 10 microns for each tissue block containing a diagnosed cervical carcinoma. DNA was extracted from the rolls using the Ionic FFPE to Pure DNA kit (Purigen Biosystems, Pleasanton, CA, USA) and quantified using a Qubit dsDNA High-Sensitivity Assay (Invitrogen, Waltham, MA, USA).

Cervical cells collected with the cervical brush were resuspended in lysis buffer and then isothermally amplified using the AmpFire HPV Screening 16/18/HR assay (Atila Biosystems, Mountain View, CA, USA). This kit qualitatively detects all hrHPV genotypes (listed above) and separately subtypes HPV 16 and 18. Samples that were positive on the screening assay were quantified using the Qubit dsDNA High-Sensitivity Assay and further genotyped as described below.

High-risk HPV genotyping was performed on the positive cervical screening samples and the 200 cervical tumor samples. Up to 200 ng of DNA in 25 μL was added to the assay tubes containing lyophilized MeltPro High Risk HPV Geno-typing Assay reagents (QuanDx/Zeesan Biotech, San Jose, CA, USA). Samples were loaded directly onto a SLAN-96 real-time PCR instrument (QuanDx/Zeesan Biotech, San Jose, CA, USA) and run using the SLAN 8.2.2 software per the manufacturer’s protocol. Each run included positive and negative internal controls provided with the kit to ensure successful amplification.

## 3. Results

We received 1994 cervical brush samples from the University of Prishtina, Kosovo. Samples were screened for hrHPV with an isothermal method as described above. Of the 1994 samples, 69 (3.5%) resulted as invalid, 1633 (81.9%) were negative, and 292 (14.6%) were positive (Figure 1). Results were considered invalid if amplification of an HPV target or the internal control was not achieved. Possible reasons included lack of DNA in the sample, challenges with sample collection, lack of full suspension in lysis buffer, sample degradation, or the presence of PCR inhibitors. All positive samples were then genotyped for hrHPV using a real-time PCR melt curve analysis as described above. Of the 292 screened positive samples, 49 (16.8%) were invalid, 23 (7.9%) were negative, and 220 (75.3%) harbored an hrHPV genotype (Figure 2). Invalid results did not have amplification of hrHPV or the internal control, and negative results did not show amplification of the hrHPV target, but did have amplification of the internal control. The genotyping assay used for this study differentiated among 14 hrHPV types. The number of positive samples and percent for each hrHPV type are shown in Table 1. HPV types 51 and 16 were found to have the highest prevalence (8.8% and 4.5%, respectively) in this patient population of screening cervical brush specimens (Figure 3). In addition, of the positive screening samples, 90% of participants further tested for genotype determination had either one or two detectable hrHPV types, 8% had three concomitant identifiable genotypes, and less than 1% were positive for four or more hrHPV genotypes. 

From a separate cohort of deidentified cases, we received 200 formalin-fixed paraffin blocks embedded with cervical cancer tissues to determine which hrHPV genotypes were most prevalent in tumors of the cervix. Of the 200 samples, 11 (5.5%) resulted as invalid, 16 (8.0%) were negative for the tested hrHPV genotypes, and 173 (86.5%) were positive for hrHPV. The breakdown and prevalence of hrHPV types in these cervical tumors is shown in Table 2. Unlike the screening samples, in this cohort of tumors, hrHPV types 16, 18, and 45 were present in 82.1%, 9.8%, and 8.7% of samples, respectively (Figure 4). Interestingly, hrHPV 51, which was most prevalent in the screening samples, was not detected in any of the neoplastic samples.

Eighty-nine percent (89%) of all cervical carcinomas harbored only one detectable hrHPV genotype, unlike the coinfection rate identified in cervical brush specimens.

## 4. Discussion

Screening strategies to proactively identify cervical intraepithelial lesions have greatly evolved since the advent of Papanicolaou screening. In recent decades, screening laboratories have adopted a more automated and uniform liquid-based cytology that incorporates testing for hrHPV in lower female genital tract tissues [23,24,25,26]. Currently, guidelines for cervical cancer screening are established based on a personalized risk-based assessment, with an algorithmic approach to treatment plans that incorporates cervical cytology, HPV DNA testing, and colposcopic biopsy results in concordance with the patient’s gynecologic history [27]. Detection of hrHPV in conjunction with, or as, a reflex test is necessary for appropriate patient management; however, few assays offer full genotyping capabilities. While improved screening methods have significantly reduced the prevalence and incidence of advanced cervical neoplasia, particularly in resource-high countries, a prophylactic vaccination is still recommended to preclude active and persistent HPV infection, ideally prior to a patient’s initial exposure [28,29].

The development and implementation of the HPV vaccine began in 2006 with the quadrivalent vaccine, protective against hrHPV (16 and 18) and lrHPV types (6 and 11, generally causing genital warts). This is the vaccine provided by GAVI. In 2014, the next generation of vaccine was introduced, protective against seven subtypes of hrHPV (16, 18, 31, 33, 45, 52, and 58) and two lrHPV types (6 and 11). The vaccine’s release paralleled the introduction of DNA-based testing for HPV. While DNA probe-based assays initially suffered from cross-reactivity and lower sensitivity, the more recent GeneXpert PCR-based assay could produce a qualitative positive or negative result for hrHPV [30,31]. Though the GeneXpert real-time PCR instrument has been successfully deployed throughout LMICs for tuberculosis testing, the intended use of this instrument for HPV testing was not as successful in providing data about population-wide hrHPV types.

Over the past two decades, numerous countries in Eastern Europe have established successful organized screening programs, with some resulting in a notable decrease in cervical cancer incidence. For example, Slovenia reported a 40% decrease after implementing a screening program in 2003 [32,33]. Some of the major challenges for creating and implementing organized screening programs included a severe lack of trained cytotechnologists, lack of centralized electronic health records, limited access to HPV genotyping, and variability in cost among countries [32,34]. One of the biggest obstacles was difficulty in operationalizing coverage to a significant percentage of the targeted population, with rates as low as 20% in some regions [32,33,34,35].

To aid in diminishing cervical cancer disparities in low-resource countries, we sought to explore the most prevalent viral genotypes that are mainly responsible for cervical neoplasia in Kosovo. For our investigative study, we utilized a multiplexed real-time PCR-based test (the QuanDx Multicolor Melting Curve Analysis assay). This assay was optimal because it provides a rapid turnaround time and high throughput (less than two hours for 96 samples) while differentiating among all 14 known hrHPV subtypes.

Furthermore, we used it to successfully screen different populations of women from various regions of Honduras to determine the prevalence of hrHPV and better predict which vaccination strategies would have optimal results [36]. In one of these published studies, we screened 1732 women in Honduras from a factory-based population of asymptomatic female employees and identified and compared the likely gaps in protection against cervical cancer had they been provided with the quadrivalent or nonavalent vaccine [36]. Even within a country as small as Honduras, with its population of nine million, our data showed geographic differences in hrHPV type between central rural Honduras compared to La Mosquitia, which is the largest wilderness area in Central America and is primarily accessible only by air and water.

Trends of increasing cervical cancer rates and mortality have been documented in Central and Eastern European countries, with an estimated overall prevalence of 10.6% of women harboring hrHPV in 8610 normal cytologies [32,37,38]. Notably, in this cohort, the most prevalent hrHPV genotypes were 16 and 18, which comprised approximately 4.0% of all cases [32,39]. Rare studies have previously examined the prevalence of HPV genotypes in Kosovo [21,22,40]. Our cohort in Kosovo showed a higher overall prevalence of hrHPV infection (14.6%) when compared to the aforementioned larger studies, which reported a hrHPV prevalence of 10.6% in the screened women of Central and Eastern Europe [32,37]. Our finding corroborates a previous study in Kosovo that found 13% of the screening cervical cytologies were positive for hrHPV [22]. That study also found a predominance of hrHPV genotypes 16, 31, and 51, which is similar to our cohort from Kosovo, that showed a prevalence of HPV genotypes 16 and 51 in 292 screening samples [22]. Large-scale studies in other Eastern European countries did not identify hrHPV 51 as one of the most common genotypes in screening samples [22,39]; however, a study in Wales identified HPV 51 as a predominant genotype in screened women [41].

Three of the top five HPV genotypes in our screened cohort comprise just ~19% of cervical cancers worldwide, giving support to a global geographic distribution of high-risk HPV genotypes. Curiously, although HPV 51 was common in our screening samples, the most common hrHPV genotypes in the cervical tumors were HPV 16, 18, and 45, consistent with global data [42,43], which may indicate less viral persistence in those infected with HPV 51. The development of cervical cancer is predicated on a persistence of infection, yet some genotypes are more easily cleared than others. One limitation of this study is not knowing the long-term outcome of screened women with positive hrHPV. Since HPV 51 is not currently covered by FDA-approved vaccines, it would be beneficial to know whether women with HPV 51 are more likely to develop neoplasia long-term. If so, these data could help support consideration for either targeted or broadened vaccine coverage and treatment guidelines based on the most prevalent genotypes in specific geographic regions of the world.

Multiple countries in Eastern Europe introduced both school-based and public health-based clinic vaccination programs in the late 2000s; however, these strategies have often encountered low coverage of the target population [32]. One of the few studies to examine vaccination rates in Kosovo found only 0.5% of women in the study had received the HPV vaccine [22]. In 2024, Kosovo established an organized vaccination program against hrHPV infection, so our data will be helpful to examine changes in overall prevalence and hrHPV genotype prevalence as more women are vaccinated [44].

Despite the advancements in HPV detection and opportunities to impact HPV-related disease progression by improved screening methods, access to these preventative resources has not significantly expanded to LMICs. While many countries in Eastern Europe have adopted cervical cancer screening programs, the overall prevalence of hrHPV in some countries, including Kosovo, remains higher than surrounding regions, perhaps due to challenges with implementing an organized screening program and limitations to vaccination. Fortunately, in the last 8 years, over 15,000 women have been screened, with 4.2% of the more than 2000 women showing atypical Pap results [45]. Since a national HPV-cotesting program has not yet been established, our results are informative for understanding the overall burden of hrHPV in screened women in Kosovo, as well as the hrHPV genotypes most identified in cervical malignancies. There are additional limitations to our study, which include a lack of demographic information on the cohort of consented patients in this study. Future studies should include this information to highlight the most at-risk age groups in Kosovo. Further, since the focus of this study was to examine the overall prevalence of hrHPV genotypes, we did not include all the demographic information about the cervical tumor samples. The distinction between tumor subtypes in relation to the hrHPV genotype coupled with the patient’s demographic information would be informative to bolster population health data and vaccination efforts. 

## 5. Conclusions

In summary, our results showed an overall prevalence of hrHPV genotypes in 14.6% of our cohort, which is similar to the estimated hrHPV prevalence of 13.1% in other published studies examining HPV infection in Kosovo. Similarly, we found hrHPV 16 to be the most prevalent high-risk genotype in both screened women and cervical tumors. Additionally, hrHPV genotypes 18 and 45 were highly prevalent in the cervical cancer specimens. The data from this study are pertinent for creating a foundation for the development and deployment of geographically targeted vaccination strategies, which could decrease the incidence of cervical cancer in regions that harbor hrHPV genotypes not found in current vaccine models. Lastly, with the new vaccination initiative in Kosovo, our pre-vaccination data will provide a helpful comparison for future studies looking at hrHPV prevalence in the post-vaccination era.

## Figures and Tables

**Figure 1 diseases-12-00189-f001:**
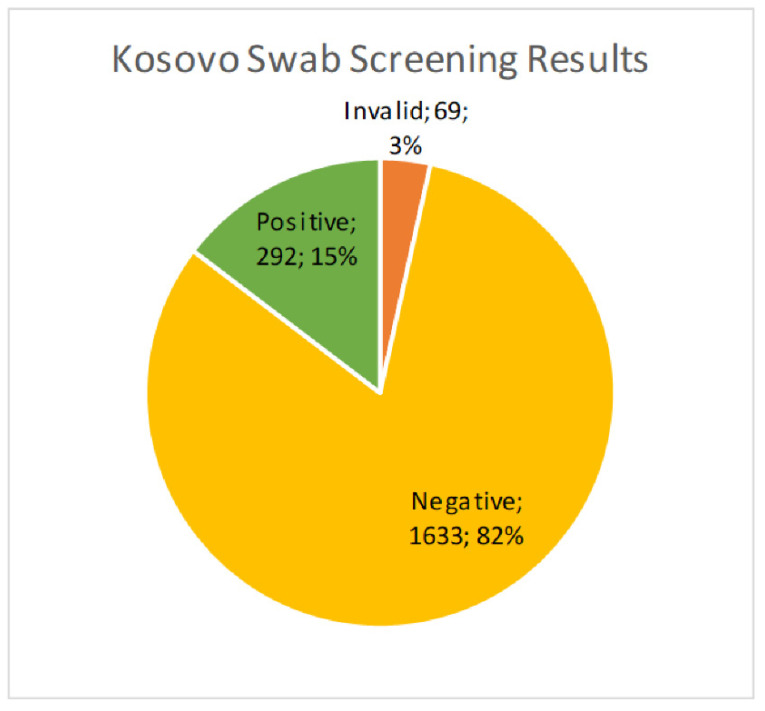
Percent of hrHPV positive cervical brush specimens used in screening a Kosovo population of healthy women, showing 15% of samples positive for hrHPV.

**Figure 2 diseases-12-00189-f002:**
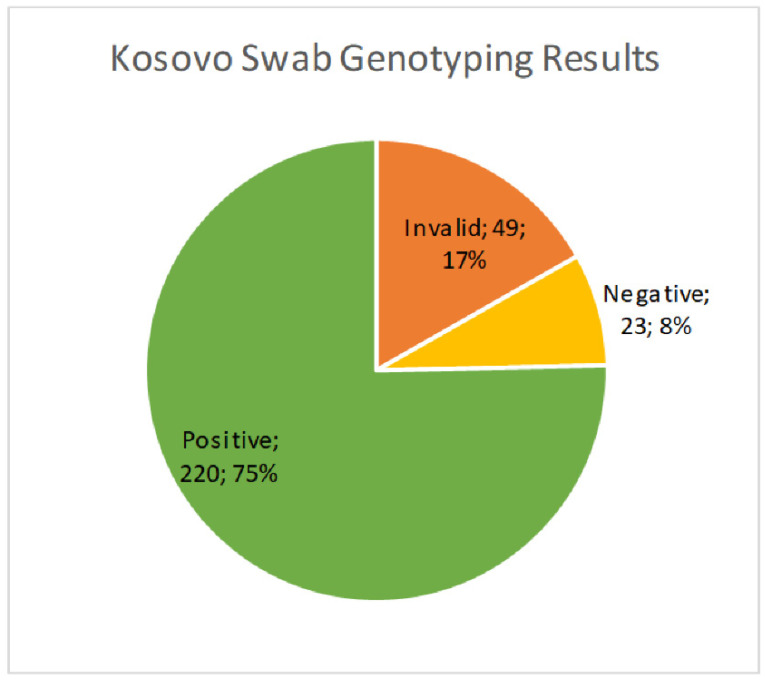
Prevalence of hrHPV positive cases from the cervical brush screening samples using a real-time PCR assay.

**Figure 3 diseases-12-00189-f003:**
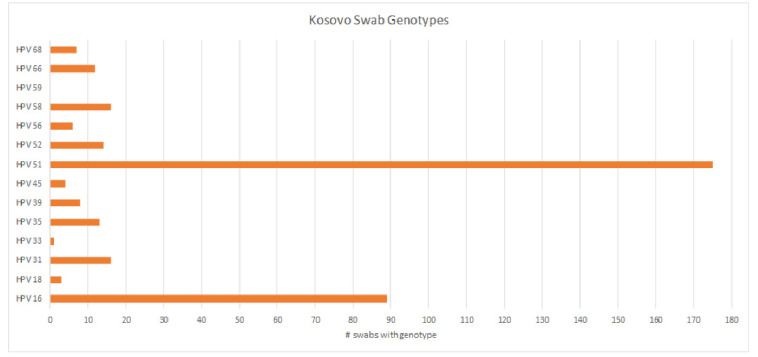
Prevalence of hrHPV genotypes among screened positive cervical brush samples, showing a high prevalence of hrHPV types 51 and 16.

**Figure 4 diseases-12-00189-f004:**
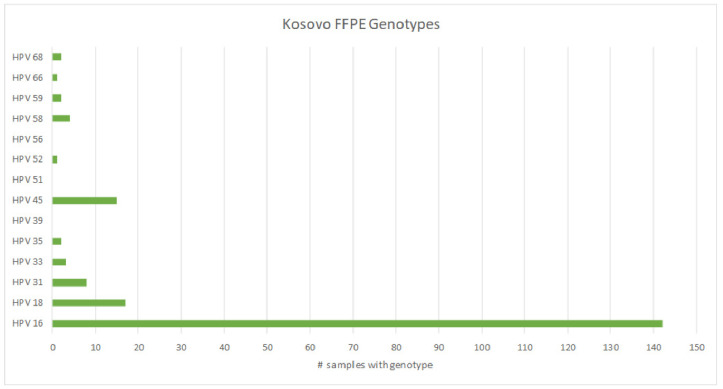
Prevalence of hrHPV types detected in DNA isolated from cervical tumors.

**Table 1 diseases-12-00189-t001:** Prevalence of hrHPV genotypes in cervical brush samples screened for hrHPV. Results are expressed as number of cases (count), percent of positive hrHPV genotyped cases, percent of total samples genotyped, and percent of total cervical brush samples screened. The coinfections row indicates the number of samples that contained more than one hrHPV genotype.

Positive Genotypes	Count	% of Genotype Positive	% of Samples Genotyped	% of Total Samples
HPV 16	89	40.5%	30.5%	4.5%
HPV 18	3	1.4%	1.0%	0.2%
HPV 31	16	7.3%	5.5%	0.8%
HPV 33	1	0.5%	0.3%	0.1%
HPV 35	13	5.9%	4.5%	0.7%
HPV 39	8	3.6%	2.7%	0.4%
HPV 45	4	1.8%	1.4%	0.2%
HPV 51	175	79.5%	59.9%	8.8%
HPV 52	14	6.4%	4.8%	0.7%
HPV 56	6	2.7%	2.1%	0.3%
HPV 58	16	7.3%	5.5%	0.8%
HPV 59	0	0.0%	0.0%	0.0%
HPV 66	12	5.5%	4.1%	0.6%
HPV 68	7	3.2%	2.4%	0.4%
Coinfections	120	54.5%	41.1%	6.0%

**Table 2 diseases-12-00189-t002:** Prevalence of hrHPV genotypes in cervical cancers. Expressed as number of cases, percent of the positive hrHPV types, and percent of total tissue cases. The coinfections row indicates the number of samples that contained more than one hrHPV genotype.

Genotype	Count	% of Genotype Positive	% of Total Samples
HPV 16	142	82.1%	71.0%
HPV 18	17	9.8%	8.5%
HPV 31	8	4.6%	4.0%
HPV 33	3	1.7%	1.5%
HPV 35	2	1.2%	1.0%
HPV 39	0	0.0%	0.0%
HPV 45	15	8.7%	7.5%
HPV 51	0	0.0%	0.0%
HPV 52	1	0.6%	0.5%
HPV 56	0	0.0%	0.0%
HPV 58	4	2.3%	2.0%
HPV 59	2	1.2%	1.0%
HPV 66	1	0.6%	0.5%
HPV 68	2	1.2%	1.0%
Coinfections	22	12.7%	11.0%

## Data Availability

Dataset available upon request from the authors.

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
