# Peer review of "High-Risk HPV Screening Initiative in Kosovo—A Way to Optimize HPV Vaccination for Cervical Cancer"

_diseases, 2024, doi:10.3390/diseases12080189_

Round 1
Reviewer 1 Report
Comments and Suggestions for Authors
new population, repetitive study design and results
Author Response
Dear Reviewer 1: Thank you for your comments and support. We appreciate you taking the time to review our manuscript.
Reviewer 2 Report
Comments and Suggestions for Authors
Well written manuscript.
1. The age, parity, sexual activity history, number of sexual partners of the women would be of great interest. The typical age groups suggested for starting HPV screening is 30-35 years, depending the age at which sexual activity starts (earlier for earlier age at first sexual contact).
2. The population of Kosovo, the percentage of women in the screenable age groups, and the way the sample size was calculated may be mentioned.
3. The incidence of cervical cancer in Kosovo may be provided.
4. It is possible serotype 51 was high in the asymptomatic women: its absence in cervical cancer smears proves that it is possibly eliminated over time
5. It is essential to formulate a gender-neutral school vaccination program for age groups 9 to 14 years.
6. It may be worthwhile to mention any previous studies from Kosovo on VIA/VILI or PAP smears to judge CIN with grades, if available
7. Is there a likelihood of following the said cohort longitudinally?
8. What treatment was offered to HPV positive women - observation and repeat HPV at 6 and 12 months, colposcopy and local ablative therapies?
Author Response
Dear Reviewer #2: Thank you for these thoughtful comments to help provide clarification to the paper. We appreciate your time in reviewing our work. Please find our responses below.
1.The age, parity, sexual activity history, number of sexual partners of the women would be of great interest. The typical age groups suggested for starting HPV screening is 30-35 years, depending the age at which sexual activity starts (earlier for earlier age at first sexual contact).
Thank you for all your feedback and comments. We agree with you that the age, parity, and sexual history would be very interesting for this study. The demographic information was not provided for this study because we were exclusively looking at hrHPV prevalence for this introductory study. We do, however, hope to continue working with our colleagues in Kosovo and expanding on this initial pilot data to incorporate additional meaningful demographic information in future publications. We have included this point into the discussion as a limitation of this study.
2. The population of Kosovo, the percentage of women in the screenable age groups, and the way the sample size was calculated may be mentioned.
Thank you for this comment.
Our sample size was determined by a statistician who worked on the Honduras project and used this as guidance to fit the number of cases within our budget/funding for this study. We do not have the data on the percentage of screenable women within the population per age group.
*We have incorporated the available information into the introduction and methods of the paper. Thank you for your kind suggestions.
3. The incidence of cervical cancer in Kosovo may be provided.
Thank you for this valuable comment. We have added references with what data is known into the paper. The overall prevalence is 13.1% and per the UNFPA, over 15,000 women were screened since 2016 and 4.2% of the cohort had atypical results. This information was added to our discussion with the associated reference.
4. It is possible serotype 51 was high in the asymptomatic women: its absence in cervical cancer smears proves that it is possibly eliminated over time.
Thank you. We agree with you that the absence of genotype 51 may be due to the immune system helping to eradicate the virus prior to evolution into cervical cancer. Thank you for helping us note this in the discussion.
5. It is essential to formulate a gender-neutral school vaccination program for age groups 9 to 14 years.
Thank you for this inquiry. The following information was shared from our colleagues in Kosovo:
From February 20th, 2024, all girls aged 12 in Kosovo are eligible to receive the human papillomavirus (HPV) vaccine for lifelong protection from cervical cancer. To comprehensively roll out the vaccine, Pristina health authorities launched a vaccination campaign at schools aiming to reach over 12 000 girls in 2024; 116 doses have already been administered during the first 2 days of the campaign. While most of these girls will be vaccinated in schools, health workers will also go door-to-door to reach girls from minority Roma, Ashkali, and Egyptian communities, who may have less contact with health and education systems. Girls in grade six across Kosovo started receiving the human papillomavirus (HPV) vaccine. According to health authorities in Kosovo it started with 15,000 units for girls aged 13.
6. It may be worthwhile to mention any previous studies from Kosovo on VIA/VILI or PAP smears to judge CIN with grades, if available
Thank you for this suggestion. We have included the available data from the following references into our article:
Fehmi Zeqiri 1, Myrvete Paçarada, Vlora Zeqiri, Gyltene Kongjeli, Pranvera Zejnullahu: Colposcopy and cytodiagnosis in the prevention of cervical malignancies. Niger J Med 2010 Oct-Dec;19(4):386-90.
Pranvera Zejnullahu Raçi 1, Fitim Raçi 2, Teuta Hadri 3: Kosovo women's knowledge and awareness of human papillomavirus (HPV) infection, HPV vaccination, and its relation to cervical cancer. BMC Womens Health. 2021 Oct 9;21(1):354. doi: 10.1186/s12905-021-01496-x.
7. Is there a likelihood of following the said cohort longitudinally?
Great question. Unfortunately, per our IRB, all of the patient information was de-identified in order to transfer it to us in New Hampshire so we are not able to follow this cohort longitudinally.
8. What treatment was offered to HPV positive women - observation and repeat HPV at 6 and 12 months, colposcopy and local ablative therapies?
Thank you for inquiring about this. We have noted this in our methods to highlight that all patients with atypical results were referred for standard treatment. HPV positive women after the primary HPV testing were recommended to repeat HPV testing after 6 -12 months. In case of the same result after the secondary HPV testing they are recommended to undertake colposcopy and biopsy. CIN2/3 women are recommended to do LEEP, Cryotherapy, Laser therapy or Conization in order to prevent progression.
Reviewer 3 Report
Comments and Suggestions for Authors
The authors of this manuscript have determined hrHPV genotypes and analyzed their frequency in screening’s cervical brush samples and cervical tumor samples in Kosovo. The work is relevant in several aspects: 1) extensive studies of this type previous had not been conducted in Kosovo and there were no data on the incidence of HPV infection and prevalence of hrHPV-related cervical neoplasia; 2) considering the WHO's global initiative to eliminate cervical cancer, it is essential to know the geographical distribution of HPV infection (especially hrHPV) to ensure effective vaccination.
Comments:
Reading the manuscript, it's unclear why the manuscript has such a title that essentially does not reflect what was written in the text. There is nothing about vaccine optimization, rather it is about optimization of vaccination (referring to WHO document). It's also unclear what the abbreviation “HPV-VOICE” means in the manuscript title, as it doesn't appear anywhere in the text.
The manuscript title should be changed to reflect the nature of the work: perhaps the “HPV Screening Initiative in Kosovo - a Way to Optimize HPV Vaccination for Cervical Cancer” or otherwise.
Lines 21 - 22 - “…the incidence of cervical carcinogenesis caused by HPV-related neoplasia” is not correct and not understandable. Need to be corrected “… the incidence of HPV-related neoplasia, cervical cancer in particular” or otherwise.
Line 32 - “cervical cancer” instead of “cervical carcinogenesis”
Line 33 - should be “highly prevalent hrHPV genotypes”
Line 53 - GAVI Alliance (GAVI)
Lines - 54 – GAVI funding
Line - 124 – “hrHPV” instead of “hrhPV.
Author Response
Dear Reviewer 3: Thank you for taking time to review our paper and to offer these great suggestions and edits.
Reading the manuscript, it's unclear why the manuscript has such a title that essentially does not reflect what was written in the text. There is nothing about vaccine optimization, rather it is about optimization of vaccination (referring to WHO document). It's also unclear what the abbreviation “HPV-VOICE” means in the manuscript title, as it doesn't appear anywhere in the text.
The manuscript title should be changed to reflect the nature of the work: perhaps the “HPV Screening Initiative in Kosovo - a Way to Optimize HPV Vaccination for Cervical Cancer” or otherwise.
- Thank you for suggesting this title which we agree is a much better representation of the manuscript. The VOICE-HPV project is a project that aims to examine the HPV prevalence in numerous countries (USA, Tanzania, Kosovo, and Honduras) as a means of comparing hrHPV genotypes from a geographic standpoint. We did not explicitly mention this background in the paper which would be confusing to readers. Your comment has helped us to see that a future paper should be written that compares and contrasts the different sites and it would be better suited to the original title. If you don't mind, we would like to use your suggested title for this paper because we think it's great!
Lines 21 - 22 - “…the incidence of cervical carcinogenesis caused by HPV-related neoplasia” is not correct and not understandable. Need to be corrected “… the incidence of HPV-related neoplasia, cervical cancer in particular” or otherwise.
We are in agreement with your edit and have made this correction in the paper.
Line 32 - “cervical cancer” instead of “cervical carcinogenesis”
We are in agreement with your suggested edit and have made this correction in the paper.
Line 33 - should be “highly prevalent hrHPV genotypes”
We are in agreement with your suggested edit and have made this correction in the paper.
Line 53 - GAVI Alliance (GAVI)
We are in agreement with your suggested edit and have made this correction in the paper.
Lines - 54 – GAVI funding
We are in agreement with your suggested edit and have made this correction in the paper.
Line - 124 – “hrHPV” instead of “hrhPV.
Thank you for noticing this typo. We have made this correction in the paper.
Reviewer 4 Report
Comments and Suggestions for Authors
Dear authors I do believe is an interesting and important article some suggestions
Methodology
· There was a sample calculation or it was prepositive?
· Did you collect samples during a specific period?
· How many clinics have participated?
· How many collogues have participated? All of them were physicians?
· There were any selection criteria for participants age?
· There was 1994 Healthy or asymptomatic participants?
· How do you enrollee participants, there was an invitation? It was in the clinic hall? please describe
· The 200 paraffin blocks have any specific carcinoma stage? They all was invasive cancer there was any specific characteristic? E.g. stage 1a1 and greater
· Is not clear where did you obtain the paraffin blocks ,they was stored previously in a bio bank? Please clarify
· Ther was a signed informed consent?
Results
Line 1,994 samples, 69 (3.5%) resulted as invalid Wy It was invalid?
Line 116 Of the 292 screened positive samples, 49 (16.8%) were invalid, 23 (7.9%) were negative. Why the samples were invalid and why they tested negative
Line 14211 (5.5%) resulted as invalid 16 (8.0%) Why they were considered invalid
I think that is important to draw some limitations of the study
And Highlight the key findings in conclusion section
Author Response
Dear Reviewer #4: Thank you for taking the time to review our paper and to offer great suggestions for improvement and clarification.
Methodology
There was a sample calculation or it was prepositive?
- Thank you for requesting this information. This study was meant to help provide baseline information for the country to help direct and support better cervical cancer awareness. Since we have previously performed these studies in Honduras, we had an approximate number that we hoped to achieve to help inform the country on hrHPV genotypes. Also, we did have a statistician assist with calculations that would insure we had enough samples but within our allotted funding. This information was added into our methods.
Did you collect samples during a specific period? How many clinics participated?
- Thank you for bringing this to our attention. The samples were collected at a single ongoing OBGyn clinic that is part of the University Clinical Center of Kosovo. We have made note of this detail in the methods section of the paper; however there was not a set time period. We continued collecting specimens until we reached our goal that was statistically significant, but still within our funding capabilities.
How many collogues have participated? All of them were physicians?
- Thank you for asking for this information to be noted. There were two professors of gynecology and obstetrics leading this process (Dr. Vlora Ademi-Ibishi & Dr. Brikena Elshani-Daci) who were helped by residents in this clinic. We included this into our methods.
There were any selection criteria for participants age?
- Thank you for requesting this information. There was not selection criteria for the participants age. This was a pilot study to examine the prevalence of hrHPV in the general population of patients who were being seen for a general gynecologic examination. We made note that the collection process was more ongoing versus in relation to specific criteria.
There was 1994 Healthy or asymptomatic participants?
- The word "healthy" was probably misleading and was exchanged for the word "asymptomatic." Thank you for bringing this to our attention. It has been corrected in the paper.
How do you enrollee participants, there was an invitation? It was in the clinic hall? please describe
- Thank you for requesting this information. The participants were not recruited. They were attending the OBGyn clinic at the University Clinical Center of Kosovo for their usual exam and were consented to participate at that time. This was clarified in the methods section.
The 200 paraffin blocks have any specific carcinoma stage? They all was invasive cancer there was any specific characteristic? E.g. stage 1a1 and greater
- Thank you for requesting this clarification. These details were not requested for the purpose of this pilot study but we hope to include all the demographic information in a future paper. This has been noted as a limitation in the discussion.
Is not clear where did you obtain the paraffin blocks, they was stored previously in a bio bank? Please clarify
- Thank you for requesting this clarification. The paraffin blocks were obtained from the archives of the Pathology Department at the University Clinical Center of Kosovo. This has been noted in our methods section.
Ther was a signed informed consent?
- Thank you for highlighting this important concern. Yes, the patients were consented prior to participation. Please see line 86.
Results
1. Line 1,994 samples, 69 (3.5%) resulted as invalid Wy It was invalid?
- Thank you for pointing out this important question. Invalid samples did not have amplification of a HPV target or the internal control. This usually suggests there is not enough DNA in the sample, which could be due to swab not properly collected, sample not fully suspended in lysis buffer, sample degradation, or presence of PCR inhibitors. This explanation was included in the results to explain our data pertaining to this comment and the third comment below from line 14211.
2. Line 116 Of the 292 screened positive samples, 49 (16.8%) were invalid, 23 (7.9%) were negative. Why the samples were invalid and why they tested negative
- Thank you for pointing out this important question. The 292 samples were positive via the screening assay (from question 1). These were then run on the genotyping assay, which is a different kit. The invalid samples did not have any amplification in hrHPV target or internal control. Negative samples did not have any amplification in hrHPV target, but did have internal control amplification. We included this explanation in the results of the paper.
3. Line 14211 (5.5%) resulted as invalid 16 (8.0%) Why they were considered invalid
-
Thank you for pointing out this important question. Invalid samples did not have amplification of a HPV target or the internal control. This usually suggests there is not enough DNA in the sample, which could be due to swab not properly collected, sample not fully suspended in lysis buffer, sample degradation, or presence of PCR inhibitors. We included this explanation in the results of the paper.
I think that is important to draw some limitations of the study
And Highlight the key findings in conclusion section
-
- Thank you for this editorial suggestion. Limitations are important to note for this study and have been added to the discussion section of the paper. We also added the key findings to the conclusion as suggested. Thank you!
Round 2
Reviewer 4 Report
Comments and Suggestions for Authors
Thak you for taking into account my sugestions.
Just one coment
Line 87 " To generate a statistically significant number 87 of cervical screening samples as determined by a statistician"
Prhaps is propistive sample of 87 cervical screning specimens? otherwise is important to explain the statistical significance
Author Response
Dear Reviewer #4: Thank you for reviewing our updated manuscript and for this additional clarification.
Thak you for taking into account my sugestions.
Just one coment
Line 87 " To generate a statistically significant number 87 of cervical screening samples as determined by a statistician"
Prhaps is propistive sample of 87 cervical screning specimens? otherwise is important to explain the statistical significance
I agree with you that the word "prospective" may be better suited for this aspect of the paper. A little background - this IRB approved study was approved for multiple regions of the world and we had to get enough samples from the different locations while still staying within the budget that was allotted. The number of samples in Kosovo was determined based on these overall guidelines.
Do you believe it is important to provide this background information on the VOICE study that I just mentioned? If you think we should include more information regarding the basis of the VOICE study, then I am happy to add more into the methods. If you think the minor edit will suffice, I have included a rewrite of that sentence in the methods section of this resubmitted paper. Thank you again for taking time to review our manuscript.